# Trauma-focused treatments for depression. A systematic review and meta-analysis

**Sarah K. Dominguez[1], Suzy J. M. A. Matthijssen[2,3], Christopher William Lee[1,4]***

**1** School of Psychology and Exercise Science, Murdoch University, Murdoch, WA, Australia, **2** Altrecht Academic Anxiety Centre, Utrecht, The Netherlands, **3** Faculty of Social Sciences, Department of Psychology, Utrecht University, Utrecht, The Netherlands, **4** Faculty of Health and Medical Sciences, The University of Western Australia, Crawley, WA, Australia

* chris.lee@uwa.edu.au

**Data Availability Statement:** The complete search strategy is detailed in the section with the subheading search strategy in the manuscript and the articles found and analysed are detailed in Table 1.

## Abstract

### Background

Trauma-focused treatments (TFTs) have demonstrated efficacy at decreasing depressive symptoms in individuals with PTSD. This systematic review and meta-analysis evaluated the effectiveness of TFTs for individuals with depression as their primary concern.

### Methods

A systematic search was conducted for RCTs published before October 2019 in Cochrane CENTRAL, Pubmed, EMBASE, PsycInfo, and additional sources. Trials examining the impact of TFTs on participants with depression were included. Trials focusing on individuals with PTSD or another mental health condition were excluded. The primary outcome was the effect size for depression diagnosis or depressive symptoms. Heterogeneity, study quality, and publication bias were also explored.

### Results

Eleven RCTs were included ($n = 567$) with ten of these using EMDR as the TFT and one using imagery rescripting. Analysis suggested these TFTs were effective in reducing depressive symptoms post-treatment with a large effect size [$d = 1.17$ (95% CI: 0.58~1.75)]. Removal of an outlier saw the effect size remain large [$d = 0.83$ (95% CI: 0.48~1.17)], while the heterogeneity decreased ($I^2 = 66\%$). Analysis of the 10 studies that used EMDR also showed a large effect [$d = 1.30$ (95% CI: 0.67~1.91)]. EMDR was superior to non trauma-focused CBT [$d = 0.66$ (95% CI: 0.31~1.02)] and analysis of EMDR and imagery rescripting studies suggest superiority over inactive control conditions [$d = 1.19$ (95% CI: 0.53~1.86)]. Analysis of follow-up data also supported the use of EMDR with this population [$d = 0.71$ (95% CI: 1.04~0.38)]. No publication bias was identified.

### Conclusions

Current evidence suggests that EMDR can be an effective treatment for depression. There were insufficient RCTs on other trauma-focused interventions to conclude whether TFTs in

**Funding:** The authors received no specific funding for this work.

**Competing interests:** All authors report receiving personal fees from private clients and income from delivering therapist training in depression and PTSD. This does not alter our adherence to PLOS ONE policies on sharing data and materials.

general were effective for treating depression. Larger studies with robust methodology using EMDR and other trauma-focused interventions are needed to build on these findings.

## Introduction

It is estimated that more than 264 million individuals worldwide suffer from depression, with the World Health Organisation stating that it is currently the leading cause of disease burden worldwide [1, 2]. Despite the pervasive nature of this disorder, and a substantial body of evidence devoted to understanding and shaping best-practice psychological interventions, approximately 40% of the individuals suffering from depression fail to respond positively to these evidence-based treatments [3]. In a meta-analysis on the effects of psychotherapies for major depressive disorder (MDD), 62% of the patients were found to no longer meet the criteria for MDD [4]. However, 43% of the participants in control conditions and 48% of people in care-as-usual conditions also no longer met the criteria for the diagnoses. This suggests the additional value of psychotherapy to be 14% [4]. Further, of the individuals who do recover, more than half are likely to relapse within several years of receiving these interventions [5]. Although these relapse rates may reflect the episodic nature of the disorder [6], further investigation into evidence-based interventions is warranted.

The causes of depression and the factors that maintain this disorder are likely to be numerous. One predisposing factor for depression repeatedly identified in the literature is the experience of aversive life events in childhood [7, 8]. Individuals who identify such events are likely to have more severe depressive symptoms and a poorer response to evidence-based treatments compared with those who do not identify a history of adversities [9]. These adversities may also be involved in perpetuating depressive symptoms. The nature of these adversities that impact later functioning is varied and subjective [10]. For a diagnosis of PTSD the adverse experience needs to involve actual or threatened injury or death or sexual assault [11, 12]. However, this restricted criterion does not allow for all adversities that cause ongoing distress or impairment [10, 13]. For example, there is evidence that other, objectively less severe events, such as bullying or neglect, also have lasting psychological impact [10, 14].

While the link between depression and adversities is well established [8, 9], how to mitigate the impact of this link in individuals with depression, in the absence of PTSD is less clear. One way to investigate this is to look at the symptoms related to these adversities that may be maintaining the depression. One such symptom is the existence of intrusive memories [15–17]. In a recent meta-analysis, adults with depression were found to experience intrusive memories more frequently than healthy controls and at a similar frequency to adults with PTSD [16]. Further, in another meta-analysis, the frequency and aversiveness of such memories were found to be associated with severity of a patient's depression [18]. There is evidence that individuals with intrusive memories often engage with problematic strategies such as avoidance or rumination on the event or related thoughts and emotions, which further maintains depressive symptoms [18–20]. Avoidance can be cognitive, behavioural or experiential, while rumination is defined as repeated, uncontrollable, self-focused negative thinking related to the past [16, 18, 21]. Both avoidance and rumination of intrusive memories are strongly correlated with depression diagnosis and severity, and poorer prognosis [17, 18, 21, 22]. Therefore, it seems plausible to target these intrusive memories and related behaviours, to enhance depression treatment [16, 18].

Trauma-focused treatments (TFTs) are a range of treatments that are primarily used to address symptoms of PTSD, including intrusive memories, related cognitions, emotions and

experiential avoidance [23]. TFTs include prolonged exposure, trauma-focused cognitive behavioural therapy (CBT), eye movement desensitisation and reprocessing (EMDR), cognitive processing therapy, exposure-based cognitive therapy, and imagery rescripting (ImRs). The impact of TFTs on depressive symptoms for individuals with PTSD has been well documented [24–27]. However, until recent years the impact of trauma therapies on depressive symptoms, in the absence of a PTSD diagnosis, has not been investigated.

An increasing number of clinical trials have been conducted investigating a range of TFTs as a treatment for depression outside a PTSD diagnosis, including exposure-based cognitive therapy [28], EMDR [29], trauma-focused-CBT [30], and ImRs [31]. Due to the number of published manuscripts relating to using a TFT for depression, narrative reviews of both EMDR [32, 33] and ImRs [34] have been published. These reviews highlight that each TFT has been found to be effective with depressed individuals in the absence of a comorbid PTSD diagnosis. However, at the time of designing the current study, there was no meta-analysis on the effectiveness of treating depression with a TFT when PTSD was not the presenting issue.

The aim of this paper was to provide a synthesised analysis of the effects of TFTs on depression to date. It was hypothesised that, for individuals with depression as their primary complaint, those who receive a TFTs would be more likely to show a decrease in depressive symptoms and increase in the likelihood of remission than those who receive a control condition. This hypothesis was assessed via a systematic review of all relevant randomised controlled trials (RCTs) and a meta-analysis of the pooled data.

## Method

### Search strategy

A comprehensive search was conducted in October 2019 in Cochrane CENTRAL; Pubmed, EMBASE and Proquest: PsycInfo, to attempt to identify all published and unpublished RCTs. No language restrictions were set so long as data and an English abstract were available. Articles of interest were identified using the following search terms: *"eye movement desenti?ation and reprocessing"* OR *"eye movement desenti?ation"* OR *"EMDR"* OR *"trauma focused treatment"* OR *"trauma focused CBT"* OR *"TF-CBT"* OR *"brief eclectic psychotherapy"* OR *"BEPP"* OR *"exposure"* OR *"cognitive processing therapy"* OR *"CPT"* OR *"narrative exposure therapy"* OR *"NET"* OR *"imagery rescripting* AND *depressi\**, *dysthymic* OR *MDD* OR *MDE*.

In addition to this search, the reference lists of relevant reviews and studies were screened. Finally, the WHO international clinical trials registry platform was screened to identify any unpublished papers that were missed. Prior to the search commencing the review was registered with PROSPERO (ID 155541).

### Selection criteria

Papers included in the analysis were based on the following PICO:

**Population.** Individuals with a primary mental health diagnosis or subclinical symptoms of major depressive disorder, persistent depressive disorder, dysthymia, premenstrual dysphoric disorder, depressive disorder due to another medical condition, other specified depressive disorder or unspecified depressive disorder. Studies that required participants to have a PTSD diagnosis were excluded from our analysis.

**Intervention.** Any treatment that was originally designed to target symptoms of posttraumatic stress disorder. This includes—but may not be limited to—EMDR; trauma-focused CBT, brief eclectic psychotherapy, (prolonged/imaginary) exposure, cognitive processing therapy, narrative exposure therapy, and ImRs.

**Comparator.** All other psychological and pharmacological treatments including standard care and waitlist.

**Outcomes.** Symptoms of depression measured with any valid instrument, taken post-treatment or at any follow-up period.

Studies were excluded if they were not a RCT, if the primary intervention was a non-trauma-focused intervention, or if the primary outcome was another mental health condition, including PTSD. As the focus of our study was individuals with depression outside a PTSD diagnosis, participants in the studies were not required to have a history of trauma as defined by the major diagnostic criterion [i. e. 11, 12]. However, studies with an inclusion criterion of adverse or traumatic experiences, that specified a broad definition of adversities to include incidents such as neglect, parental divorce, or bullying, were included in the analysis.

Using the software program Covidence (https://www.covidence.org/) to store and manage the identified studies, two authors (SD & SM) independently conducted the manual review of the papers using the above PICO, and the stated inclusion and exclusion criteria. Any discrepancies in the screening were discussed with a third researcher (CL) until consensus was reached. The data was then extracted from the included studies independently by two members of the research team (SD & SM).

## Quality assessment

The quality of the included studies was evaluated using the Cochrane Collaboration's risk of bias for randomised studies tool [35]. As it is not possible for those delivering or receiving the intervention to be blind to the treatment, three other criteria were also assessed, as has been done by other researchers in other reviews of psychological interventions [36]. These criteria evaluated therapeutic allegiance, treatment fidelity and therapist qualifications. Studies were rated as low or high risk of bias or some concern. Assessment was conducted independently by two members of the research team (CL and SD). Any discrepancies were discussed together with the third author (SM) until a consensus was met.

## Meta-analytic procedure

The impact of the TFTs on depressive symptoms was compared with all control conditions, including active psychological treatments and waitlist or non-psychological care. A pooled controlled condition has been used in other meta-analyses looking at depression across varied populations [36, 37]. Analysis of variable change from baseline was conducted using post-treatment and follow-up results, from continuous and dichotomous outcomes. For studies that involved multiple variables, such as active control and inactive control, or two measures of depression, the mean of both variables was used unless otherwise stated.

The software program Comprehensive Meta Analysis (Version 3) was used to calculate the pooled effect size of all relevant studies [38]. Effect sizes (Cohen's d) of .8 can be considered large, .5 moderate and .2 small [39]. The studies were expected to vary in terms of number of sessions, type of TFT and the clinical severity of the participants; therefore, a random-effects model was used in all analyses [40]. An exception to this was if the number of studies was less than five, in which case fixed effects were used in line with current recommendations [40]. Heterogeneity was assessed using the $i^2$ statistic in which 25% refers to low, 50% to moderate and 75% to high heterogeneity [41, 42].

## Subgroup and sensitivity analysis

Subgroup analyses were conducted to see if there was a difference between active and inactive control conditions, and to investigate the efficacy of TFTs when delivered as a standalone

therapy (with no other psychological intervention) and as an adjunct to other psychotherapy. In addition, sensitivity analyses were conducted, removing any significant outliers. As EMDR was the TFT used in all but one study, an additional analysis was conducted, including only EMDR studies. A baseline to follow-up analysis was also conducted on studies where follow-up data was available.

### Publication bias

The likelihood of a publication bias was assessed using Egger's regression intercept test and a funnel plot. As no publication bias was detected Duval and Tweedies trim and fill procedure was not used.

## Results

### Search results

As shown in Fig 1, 751 studies were identified in the initial search, including 372 duplicates which were removed. Abstracts and titles of the remaining 379 studies were screened. This resulted in the exclusion of 354 papers. Studies that were excluded were tagged with the reason for exclusion. Studies were excluded primarily if a non-randomised trial design was used, the treatment provided was not a TFT, the participants in the study were required to have a PTSD diagnosis or a related traumatic experience, a primary diagnosis of another mental health disorder was an inclusion criterion, and finally, if complete study data could not be obtained. Data was considered as unavailable only after an attempt to contact the study authors did not result in the data being obtained. This was the case for two studies. In one study, the data was not accessible, as it was not available in English [43]. In another, results could not be sourced, despite being recorded in the trial register [44]. The majority of papers ($n = 304$) were excluded because PTSD was the primary focus or an inclusion criterion. The full text screening of the remaining papers ($n = 25$) was conducted, which resulted in 14 papers being excluded (see Fig 1). Following this, 11 studies, which involved a total of 567 participants, were considered eligible and included in the analysis.

### Characteristics of included studies

Eight studies were published in peer-reviewed journals and three were unpublished dissertation theses, with one of these being published after the search was conducted. One unpublished paper was written in 2001, and the rest of the studies were published or written between 2015 to 2020.

All of the studies included adult participants. As shown in Table 1, seven were conducted on depressed patients in general, three studies were conducted on patients with comorbid medical difficulties, and one study treated individuals who were caregivers of an individual with dementia. Three studies had the TFT delivered as an adjunct to other psychological treatments, while the remaining eight received the intervention as a standalone psychological treatment. With the exception of one study [57], which examined a group intervention, all participants in the other included studies received their intervention individually. One study investigated the effectiveness of self-guided ImRs in a brief and long form compared with a waitlist [31], and the remaining studies used EMDR as the TFT. In the ImRs study, the intervention was self-administered, and the number of therapy sessions was not recorded. For the remaining trials number of TFT sessions in the studies' designs ranged from 1 to 18 (average 6.5) and were between 45 to 120 minutes in duration. All studies used self-report measurements and one also used a structured clinical interview. The most common outcome measure

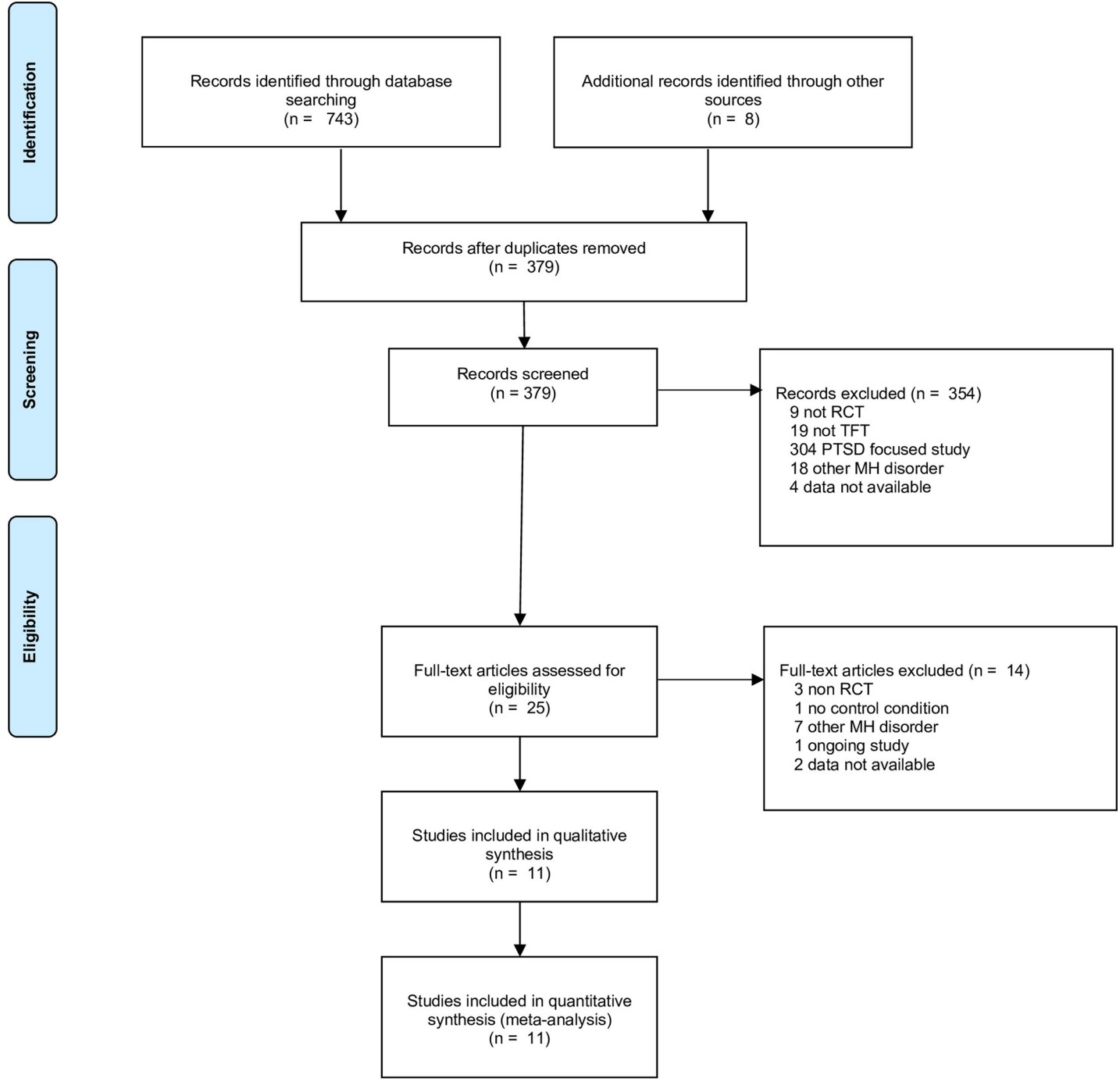

**Fig 1. Flow chart of inclusion of trials for the meta-analysis of studies on trauma-focused treatments and depression.**

used was the Beck Depression Inventory, Second Edition (BDI II) (seven studies). Of the 11 papers, four trials compared a TFT to an active psychological intervention and eight compared it to an inactive, or non-psychological intervention. One study had both active and inactive control conditions. Four studies measured depressive symptoms at follow-up periods ranging from 1 to 6 months post-treatment.

**Table 1. Description of included studies.**

| Study | Interventions | n | TFT sessions and delivery | Population | Target | Measure | Time point | Dropouts | Overall risk of bias |
|---|---|---|---|---|---|---|---|---|---|
| Behnammoghadam et al., 2015 [45] | EMDR vs UC | 60 | 3 x 45–90 min on alternate days; standalone | Cardiac patients with BDI II score >17, Iran | Experiences relating to cardiac arrests | BDI II | Post | Not reported | High risk |
| Dominguez et al., 2020 [46] | TAU + EMDR vs assertiveness (CBT) + TAU vs TAU | 49 | 3 x 90 min; adjunct | Clinical and subclinical depression, Australia | Past aversive events, episodic in nature and thematically linked to current symptoms | SCID 5; DASS 42 | Post; 6 and 12 weeks | 9% | Low risk |
| Gauhar & Wajid, 2016 [54] | EMDR vs WL | 17 | 6–8 weekly session 60 min; standalone | Clinical MDD diagnosis (DSM IV TR), Pakistan | Past aversive events, episodic in nature and thematically linked to current symptoms | BDI II | Post | 35% | Some risk |
| Hase et al., 2018 [47] | EMDR + TAU vs TAU | 30 | 4–12 (1–2 per week); adjunct | Psychiatric inpatients (diagnostic interview and BDI-II >12), Germany | Past aversive events, episodic in nature and thematically linked to current symptoms | BDI II SCL 90-R | Post | Not reported | High risk |
| Hogan, 2001 [48] | EMDR + TAU vs CBT +TAU | 30 | 1 x 60 min; adjunct | Mood disorder or adjustment disorder with depressed mood, USA | Past aversive events, episodic in nature and thematically linked to current symptoms | BDI II | Post | Not reported | High risk |
| Kao et al., 2018 [55] | EMDR vs UC | 57 | 4 x 60–90 min weekly; standalone | Patients with heart failure, Taiwan | Most unpleasant experience of heart failure | BDI II | Post; 1 & 3 months | 9% | High risk |
| Moritz et al., 2018 [31] | Self guided ImRs (brief and long form) vs WL | 127 | Self administered over 6 weeks; standalone | Clinical, Germany | Past aversive events, episodic in nature and thematically linked to current symptoms | BDI II | Post | 21% | Some risk |
| Study | Interventions | n | TFT sessions and delivery | Population | Target | Measure | Time point | Dropouts | Study quality |
| Ostacoli et al., 2018 [56] | EMDR vs CBT | 66 | 15 +/-3; standalones | Treatment resistant depression (BDI >13 and MINI), Italy and Spain | Past aversive events, episodic in nature and thematically linked to current symptoms | BDI II | Post; 6 months | 20% | Some risk |
| Passoni et al., 2018 [57] | EMDR vs WL (delayed treatment) | 33 | 8 group x 120 min over two months; standalone | Primary Carers of dementia patients, Italy | Issues related to caring for dementia | AD-R | Post | 25% | High risk |
| Rahimi et al., 2018 [58] | EMDR vs UC | 90 | 6 x 30–45 min 3 sessions per week; standalone | Patients receiving haemodialysis with HADS score in borderline or clinical range, Iran | Traumatic haemodialysis scene | HADS | Post | 0% | High risk |
| Su, 2018 [59] | EMDR vs CBT | 8 | 10; standalone | Diagnosis of depression, USA | Past aversive events, episodic in nature and thematically linked to current symptoms | PHQ-9 | 1 month | 0% | Some risk |

*n* = number analysed; EMDR = eye movement desensitisation and reprocessing; UC = usual care; BDI II = Beck Depression Inventory, Second Edition; TAU = treatment as usual; CBT = non trauma-focused cognitive behavioural therapy; SCID 5 = Structured Clinical Interview for DSM 5; DASS 42; Depression, Anxiety and Stress Scale– 42; ITT = intent to treat; WL = waitlist; MDD = major depressive disorder; DSM IV = Diagnostic and Statistical Manual, Fourth Edition; SCL 90-R Symptom Checklist–90-Revised; ImRs = imagery rescripting; MINI = Mini-International Neuropsychiatric Interview; AD-R = Anxiety and Depression Scale–Reduced Form; HADS = Hospital Anxiety and Depression Scale; PHQ-9 = Patient Health Questionnaire– 9.

**Table 2. Quality assessment: Risk of bias.**

| | Behnammoghadam et al., 2015 [45] | Dominguez et al., 2020 [46] | Gauhar & Wajid, 2016 [54] | Hase et al, 2018 [47] | Hogan, 2001 [48] | Kao et al., 2018 [55] | Moritz et al., 2018 [31] | Ostacoli et al., 2018 [56] | Passoni, 2018 [57] | Rahimi, 2018 [58] | Su, 2018 [59] |
|---|---|---|---|---|---|---|---|---|---|---|---|
| Bias arising from the randomisation process | - | + | ? | + | ? | ? | + | + | ? | ? | ? |
| Bias due to deviations from the intended interventions | - | ? | ? | ? | - | ? | ? | ? | ? | ? | ? |
| Bias due to missing outcome data | - | + | ? | - | - | ? | + | ? | + | + | + |
| Bias in the measurement of the outcome | ? | + | ? | ? | ? | ? | ? | ? | ? | ? | ? |
| Bias in the selection of the reported results | + | + | + | + | + | + | + | + | + | + | + |
| Therapist allegiance | - | + | - | - | ? | - | NA | ? | - | - | ? |
| Treatment fidelity | - | + | + | - | ? | ? | ? | ? | - | - | + |
| Therapist qualifications | - | + | + | - | ? | - | NA | + | - | - | ? |
| Overall bias | - | + | ? | - | - | - | ? | ? | - | - | ? |

+ = low risk of bias;— = high risk of bias;? = some risk of bias; NA = Not applicable due to self-administered intervention.

## Quality of included trials

Results of the quality analysis are presented in Tables 1 and 2. Of the 11 studies, one was rated as low risk of bias, six were rated as being high, and four were rated at some risk. All studies reported at least some risk of bias with regard to deviations from the intended interventions as neither the therapist nor client could be blind to the treatment condition. Further, all but one study was rated as some risk bias regarding the measurement of the outcome as they relied only on self-report data. Dropout numbers ranged from 0% to 35%, with two studies not reporting the attrition rates.

## Effect of TFTs at post-assessment

Summary results of all meta-analyses are shown in Table 3. Of the 11 studies, one did not report post-treatment data (follow-up data only). The remaining ten studies investigated the difference between a TFT and a control condition post-treatment totaling 559 participants. Two studies had three treatment groups, and two used two measures to assess depressive symptoms. As shown in Fig 2. the mean effect size was large [$d = 1.17$ (95% CI: 0.58~ 1.75)].

**Table 3. Meta-analysis of studies comparing the effects of trauma-focused treatments on depression.**

| Study | | $N_{analysed}$ | D | 95% CI | $I^2$ |
|---|---|---|---|---|---|
| Post analysis | | | | | |
| Studies with EMDR only | Random effects | 432 | 1.29 | 0.67~1.91 | 87.60 |
| Studies with EMDR only without outlier | Random effects | 372 | 0.93 | 0.61~ 1.25 | 50.33 |
| Active control comparison (CBT) only | Fixed effects | 129 | 0.66 | 0.31~ 1.02 | 0 |
| Follow-up data | Fixed effects | 180 | 0.71 | 0.38~ 1.04 | 0 |

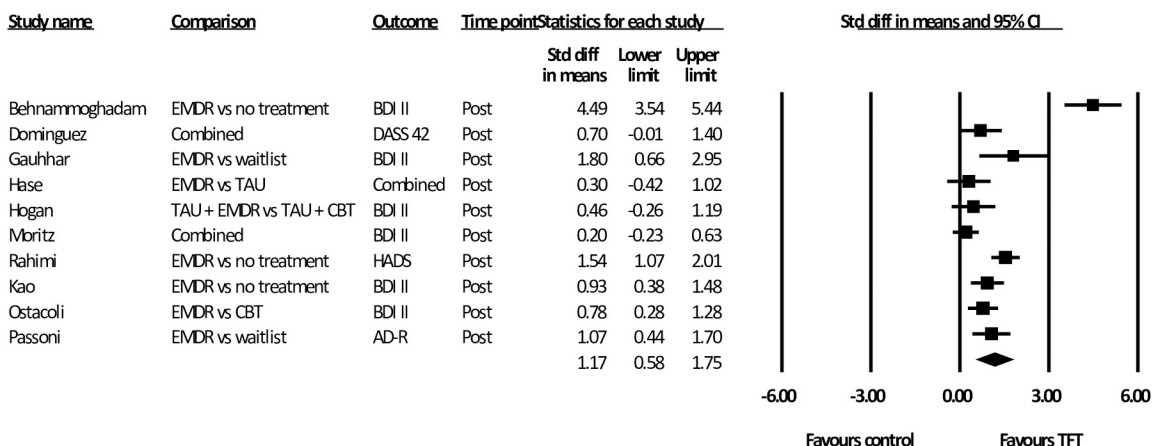

**Fig 2. Standardised effect sizes of trauma-focused treatments for depression compared to control conditions at post-test.**

Heterogeneity was high $(i^2 = 88.65)$. As one study appeared to be a significant outlier [45] we also conducted the analysis with this study removed ($n = 499$). This resulted in an improvement in heterogeneity ($i^2 = 66.24$) and precision (95% CI: 0.48~ 1.17). While effect size decreased, it was still large ($d = 0.83$).

Further analysis examined the post-assessment results for the nine studies that included only EMDR and a non-trauma focused control condition ($n = 432$). Analysis showed a large effect size [$d = 1.29$ (95% CI: 0.67 ~ 1.91)], and high heterogeneity ($i^2 = 87.60$). When the outlier was removed ($n = 372$) heterogeneity was moderate ($i^2 = 50.33$) and the effect size was still large [$d = 0.93$ (95% CI: 0.61~ 1.25)].

## Subgroup analysis at post-assessment

The first subgroup analysis compared active and inactive control conditions (Fig 3). Due to one study having both an active and inactive control, multiple treatment groups were considered independently for the purpose of this analysis. All three trials that compared a TFT to an active control used EMDR as the TFT and CBT (non-trauma-focused) as the active control intervention ($n = 129$). The mean effect size of this analysis was moderate [$d = 0.66$ (95% CI: 0.31~ 1.02)] with zero heterogeneity in favour of the TFT. Of the eight studies ($n = 446$) that

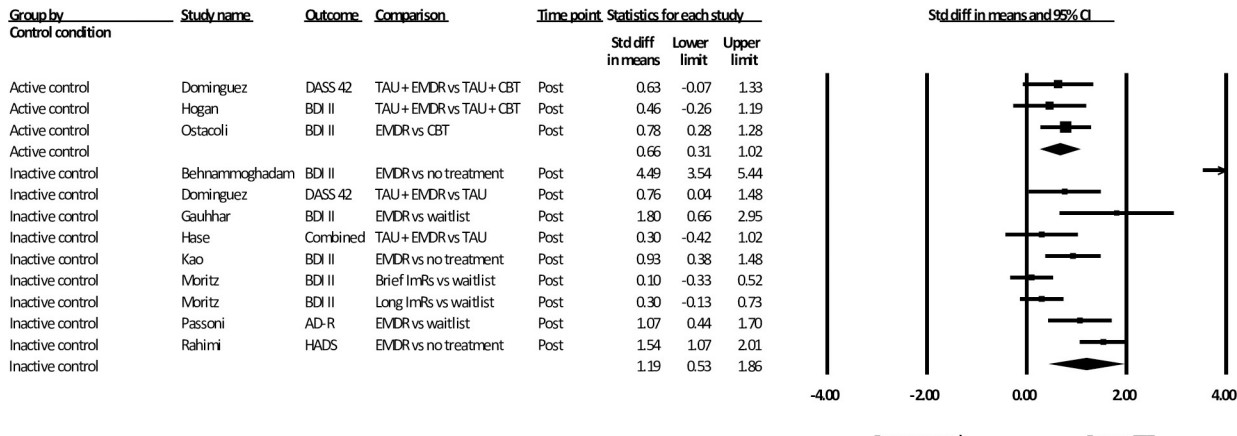

**Fig 3. Subgroup analysis: Standardised effect sizes of trauma-focused treatments for depression compared to active and inactive control conditions.**

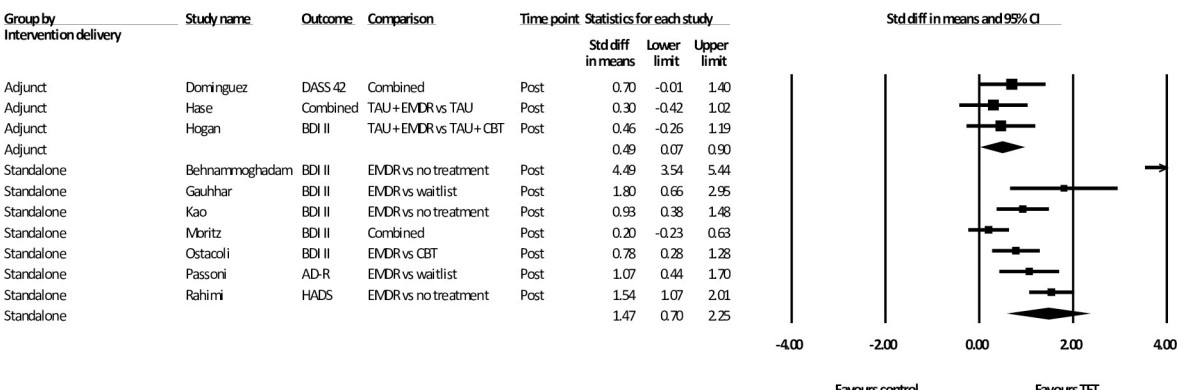

**Fig 4. Subgroup analysis: Standardised effect sizes of trauma-focused treatments for depression delivered as an adjunct or standalone therapy, compared to control conditions.**

compared a TFT to inactive control, one used ImRs as the TFT and the remaining seven used EMDR. The mean effect size was large [$d = 1.19$ (95% CI: 0.53~ 1.86)] and heterogeneity was considerable ($i^2 = 90.95$). Analysis with the identified outlier removed resulted in an improvement in heterogeneity ($i^2 = 73.48$) and precision (95% CI: 0.62~ 1.06). While the effect size decreased, it was still large ($d = 0.84$).

Analyses of intervention delivery comparing standalone and adjunct treatments are presented in Fig 4. Three studies ($n = 109$) delivered the TFT as an adjunct to other psychotherapy, with EMDR being the TFT used in all trials. The adjunct psychotherapy used in two of these trials was delivered in groups and based on a CBT or psychodynamic model [46, 47], and one involved individual CBT [48]. Analysis of the results of these studies revealed in a moderate effect size [d = .49 (95% CI: 0.07~ 0.90)] with zero heterogeneity. Analysis of the TFT as a standalone intervention ($n = 450$) showed a larger effect size [$d = 1.47$ (95% CI: 0.70~ 2.25)] as well as an increase in heterogeneity ($i^2 = 91.75$).

## Effects at follow-up

As shown in Table 2, four studies measured a total of 180 participants at follow-up, and all used EMDR as the TFT. For the two studies that used multiple follow-up assessments, analysis was conducted using the mean of these time points. The mean follow-up occurred at 11 weeks. The effect at follow-up was moderate [$d = 0.71$ (95% CI: 0.38~ 1.04)] with zero heterogeneity (Fig 5). The analysis was also calculated using the last follow-up measure in these studies. This

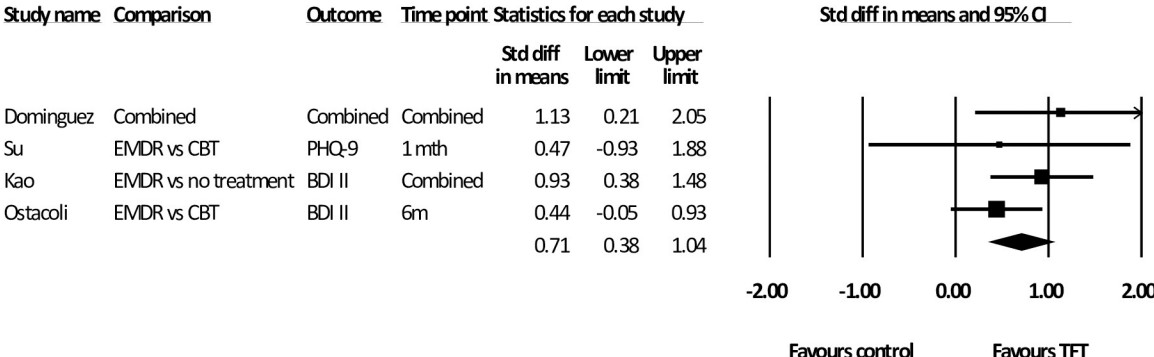

**Fig 5. Standardised effect sizes of trauma-focused treatments for depression compared to control conditions at follow-up.**

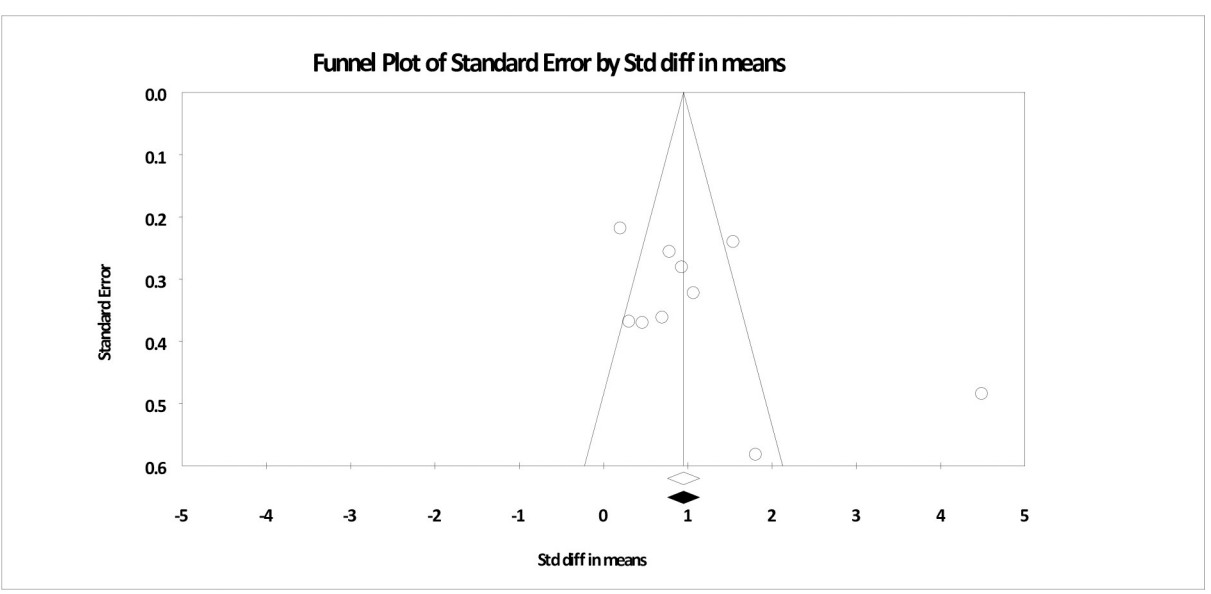

**Fig 6. Funnel plot based for main analysis of trauma-focused treatments for depression compared to control conditions at post-test.**

resulted in a mean time of 13 weeks and the results were similar [$d$ = .71 (95% CI: 0.38~1.05), ($i^2$ = 0)].

## Publication bias

Publication bias was evaluated for the main analysis. As shown in Fig 6, visual analysis of the funnel plot highlights that three studies reported effect sizes outside the expected area, with two larger than expected and one smaller. One study with a larger effect size had been identified earlier as an outlier [45]. As discussed above, the removal of this study did not significantly alter the outcome of the analysis. An analysis using Egger's regression intercept approach indicated no significant evidence of publication bias ($t$ = 1.54; $p$ = .16).

## Discussion

The current meta-analysis was conducted to assess the effect of trauma-focused treatments (TFTs) for individuals with symptoms of depression as a primary presenting concern. Of the 11 studies identified to meet the inclusion criteria, ten used eye movement desensitisation and reprocessing (EMDR) as the TFT and one used imagery rescripting (ImRs). Therefore, caution should be exercised in generalising the results of this paper for interventions other than EMDR. The results of analysis of all eleven studies support the use of these interventions for individuals with depression. Data show a large effect size when looking at pre-post comparisons across ten studies (EMDR and ImRs) when compared with any other (active or non-active) condition, even when a significant outlier was removed. Analysis of the EMDR studies on pre-post treatment change for individuals with depression again shows a large effect with or without an outlier included in the analysis. Moreover, it appears that the effects of EMDR on depression measures are sustained even after treatment has ended. Four studies evaluated the ongoing impact of EMDR in follow-up periods from 1 to 6 months. The treatment effect size was moderate for this period of time. Analysis of the funnel plot was not consistent with any significant publication bias.

Based on eight studies, the effect size was large when TFTs (EMDR and ImRs) were compared with an inactive control. Of the three studies that used an active control, all used EMDR for the TFT and non-trauma-focused CBT as the active control. Analysis of EMDR compared with CBT showed that EMDR was more likely to decrease depressive symptoms than CBT post-treatment, with a moderate effect. This is important as, although CBT is recommended as the first-line psychological treatment for depression, it does not result in the desired symptom reduction for all individuals. Therefore, the evidence that there is an additional psychological intervention that is at least as effective as CBT but possibly targets different mechanisms or maintaining factors, such as intrusive memories, is welcomed.

Three studies investigated the effect off TFTs (EMDR) when delivered as an adjunct to other psychotherapy, and the remainder investigated the effect of TFTs (EMDR and ImRs) as a standalone psychological intervention. Both modes of intervention were shown to be effective in decreasing depressive symptoms when compared with control conditions with moderate and large effect size, respectively.

One of the core implications of this study is to underscore the importance of broadening of the definition of an event that is traumatic beyond the type of trauma needed for PTSD, and the use of TFTs to target these adversities. This includes experiences that are considered less severe such as bullying or relationship breakups, or adversities of omission, such as neglect. In the majority of the studies, the authors reported that the memories chosen to treat were subjectively distressing for the participant and linked thematically to the participants' current difficulties in terms of either content, affect or cognitions. The focus on adverse events as aetiologically related to current psychopathology is central to schema therapy [49]. In schema theory the experiences of having core needs not met are viewed as the basis of what Young [49] called early maladaptive schemas that then cause dysfunction throughout the person's life. In schema therapy, these adverse experiences are targeted with interventions such as ImRs, chair work, or EMDR [50].

The paucity of RCTs using interventions other than EMDR was surprising to the authors. There was one paper that was initially included in the study that examined the efficacy of TF CBT vs EMDR; however, this was excluded as there was no non trauma-focused control condition [30]. This deficit is particularly unexpected due to the diverse body of literature supporting other TFTs [28, 34, 51], including a recent book dedicated to the practise of imagery and ImRs for treating depression and bipolar disorder [52]. Accordingly, further RCTs, including multi-arm head to head studies comparing various TFTs with a control condition, are needed. In addition to efficacy studies, process studies on how different components of other TFTs, such as cognitive restructuring or behavioural change, effect variables found to mediate depression severity such as avoidance and rumination [18] could further build on our ability to understand the aetiology and maintenance of depressive disorders.

The heterogeneity of several analyses was large in this study. This is consistent with other meta-analyses of treatments for depression [53]. These might be due to the methodological variation mentioned above or the diversity of presentations that can occur in individuals with depressive symptoms. Further, this may reflect the diversity in intervention delivery and control conditions. However, despite this diversity, in some of the analyses, the heterogeneity was satisfactory. For example, after removing outliers, the effect of EMDR compared with a non-trauma-focussed controlled condition was large, even though heterogeneity was moderate. This was somewhat surprising given that the EMDR treatment protocols used across these studies varied, as were the session duration and intensity, and the nature of the samples. Thus, care needs to be taken when interpreting these findings, especially as the results presented in these analyses are mainly based on self-report data. Accordingly, it would be useful to conduct

further research with multi-site studies, using similar protocols, clinician administered assessment and similar samples to increase the certainty of these findings.

The current study highlighted the substantial symptom reduction experienced for people with depression diagnosis or depressive symptoms following TFTs targeting negative memories related to current symptomology. Of the studies included in this analysis none involved a formal assessment for PTSD to allow for the exclusion of those with a comorbid diagnosis. Although most studies that described the memories targeted in the TFT treatment condition would not meet the PTSD severity criteria, future studies assessing and excluding individuals with a PTSD diagnosis are needed to increase the confidence in the assertion that the TFT can be effective in decreasing mood outside a PTSD diagnosis.

There are several other limitations in the current study. In addition to only one non-EMDR study, only one paper evaluated two different active treatment modes. Therefore, the generalisability of the conclusions in the meta-analyses beyond EMDR or the comparative efficacy of TFTs remains to be established. Most studies used inactive controls, and those that did use active controls only used non-trauma-focused CBT. Accordingly, the added advantage of a TFT in general beyond non-trauma-focused CBT for depression cannot be ascertained from this study.

Although we limited our review to studies that used RCTs, there was a diversity in methodical rigour in the designs of the included studies, with several studies rated as having a high risk of bias. For example, several of the studies in the analysis involved relatively small samples. Further, only one study used an observer-rated clinical interview to measure change in depression diagnosis, the other ten assessed symptom change with self-report inventories. Future high-quality studies evaluating the efficacy of a range of different TFTs compared with a range of controls with outcomes, including structured clinical interviews, are needed.

To conclude, the results of this meta-analysis support the use of EMDR as a promising approach for treating depressive symptoms. There were not enough RCTs to make the same recommendation about TFTs in general. Further studies looking at a range of TFTs (including EMDR), with increased methodological rigour and larger sample sizes, would increase the confidence in these conclusions.

## Supporting information

**S1 Checklist. PRISMA 2009 checklist.**
(DOC)

## Acknowledgments

The authors would like to thank the authors of the cited papers for responding to their queries.

## Author Contributions

**Conceptualization:** Sarah K. Dominguez, Christopher William Lee.

**Data curation:** Sarah K. Dominguez, Suzy J. M. A. Matthijssen, Christopher William Lee.

**Formal analysis:** Sarah K. Dominguez, Suzy J. M. A. Matthijssen, Christopher William Lee.

**Investigation:** Sarah K. Dominguez, Suzy J. M. A. Matthijssen, Christopher William Lee.

**Methodology:** Sarah K. Dominguez, Suzy J. M. A. Matthijssen, Christopher William Lee.

**Project administration:** Sarah K. Dominguez, Suzy J. M. A. Matthijssen, Christopher William Lee.

**Resources:** Sarah K. Dominguez, Suzy J. M. A. Matthijssen, Christopher William Lee.

**Software:** Sarah K. Dominguez, Suzy J. M. A. Matthijssen.

**Supervision:** Christopher William Lee.

**Validation:** Sarah K. Dominguez, Suzy J. M. A. Matthijssen, Christopher William Lee.

**Writing – original draft:** Sarah K. Dominguez.

**Writing – review & editing:** Sarah K. Dominguez, Suzy J. M. A. Matthijssen, Christopher William Lee.

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
