## [Decision Letter · Decision Letter 0]

17 Feb 2021

PONE-D-20-24791

Trauma Focused Treatments for Depression.  A Systematic Review and Meta-Analysis

PLOS ONE

Dear Dr. Lee,

Thank you for submitting your manuscript to PLOS ONE. After careful consideration, we feel that it has merit but does not fully meet PLOS ONE’s publication criteria as it currently stands. Therefore, we invite you to submit a revised version of the manuscript that addresses the points raised during the review process.

Specifically, together with the other comments, address the concerns of reviewer 1 who highlights major changes needed to the text, 

We look forward to receiving your revised manuscript.

Kind regards,

Andrea Martinuzzi

Academic Editor

PLOS ONE

Journal Requirements:

"We have read the journal's policy and the authors of this manuscript have the following competing interests:

All authors report receiving personal fees from private clients and income from delivering therapist training in depression and PTSD."

Reviewers' comments:

Reviewer's Responses to Questions

**Comments to the Author**

1. Is the manuscript technically sound, and do the data support the conclusions?

Reviewer #1: Partly

Reviewer #2: Partly

2. Has the statistical analysis been performed appropriately and rigorously? 

Reviewer #1: Yes

Reviewer #2: Yes

3. Have the authors made all data underlying the findings in their manuscript fully available?

Reviewer #1: Yes

Reviewer #2: Yes

4. Is the manuscript presented in an intelligible fashion and written in standard English?

Reviewer #1: Yes

Reviewer #2: Yes

5. Review Comments to the Author

Reviewer #1: Thank you for the opportunity to review this manuscript reviewing trauma-focused treatments for depression. While I think the intention of the review would be of interest to readers, the frame of "trauma-focused treatments" seems misleading (discussed below). The authors conclude that EMDR would be a beneficial treatment for depression. This should be emphasized and conjectures about the use of TFTs more broadly should be removed, especially in the discussion section. Generally, highlighting in the introduction mechanisms of change underlying various TFTs will be important to add, as this provides theoretical rationale for why these treatments might be beneficial for individuals suffering from depression. In the discussion, discussing why EMDR is the only TFT used for depression in the studies reviewed is also important to examine further. Overall, this paper makes a contribution to the field, but significant changes/additions to the introduction and discussion are needed.

Introduction

Please provide peer-reviewed citations for the benefits of trauma-focused treatments for depression. Citing ISTSS does not support the suggestion that benefits of TFT for depression are well-documented.

Please fix this sentence accordingly: “An increasing number of clinical trials have been conducted investigating a range of TFTs (exposure-based cognitive therapy, EMDR, trauma-focused-CBT, and imagery rescripting) as treatments for depression.”

The following sentence is wordy and difficult to follow – please revise to something like, “Following the increase in published studies examining TFT for depression, a number of narrative reviews have been published highlighting the promise of such interventions.”

The introduction would benefit from more detailed examination of previous studies looking at TFTs for depression. What are the mechanisms of change seen across studies with different treatments? Moreover, while adverse childhood experiences are associated with depression, what factors lead to a primary diagnosis of depression over PTSD in these cases and how might this account for benefits from TFT treatment for depressed individuals? If individuals suffering from depression did not experience a significant adverse event precipitating symptom onset, would they derive the same benefits from TFTs? The intro is substantially lacking and more examination of these questions is warranted.

Method

Were individuals with co-morbid diagnoses of PTSD and depression considered?

This sentence is wordy and difficult to follow. Please consider revising…“Studies with an inclusion criterion of adverse or traumatic events, including neglect or other incidents that would fall outside of the DSM 5 diagnostic criteria for PTSD, were not excluded from the analysis.”

Please fix this sentence accordingly: “Analysis from baseline was conducted using post-treatment and at follow-up results, from continuous and dichotomous outcomes.”

Results

Please fix this sentence accordingly: “As shown in Fig 1, 751 studies were initially identified, and 372 duplicates were removed.”

Please fix this sentence accordingly: “A total of 11 studies with a sum of 567 participants were eligible for analysis.”

Is there utility in keeping the study investigating rescripting given that the remaining studies used EMDR? Moreover, given that all studies (with one exception) used EMDR rather than other trauma-focused treatments, is it possible that this speaks to the fact that trauma treatments are not necessarily beneficial for depression but treatments aimed at alleviating distress/processing negative memories are beneficial. It seems that other trauma therapies (i.e., CPT, exposure therapy, TF-CBT) emphasize exposure/cognitive restructuring/behavioral change whereas EMDR is focused on memory reconsolidation.

Was the analysis of EMDR vs. CBT looking specifically at behavioral activation or a general CBT model? Given that BA is an intended treatment for depression, this would be an important distinction.

Discussion

The general conclusion drawn from this meta-analysis is misleading – TFTs are not necessarily a promising approach to treating depressive symptoms. EMDR may be a promising approach, according to findings. Overall, the discussion would benefit from exploration of mechanisms of change underlying EMDR for depression, and some mention of why other TFT have not been used solely for depression.

General

Please read through the manuscript from grammatical errors.

Reviewer #2: This manuscript aims to provide a meta-analysis of studies using trauma-focused treatments to treat depression without a diagnosis of PTSD. Given that there are now 11 such studies it is worthwhile to summarize them. It seems to me that this paper would be more helpful with a bit more of a critique of the existing literature and attention to the question of why such treatments may be helpful.

Introduction:

The introduction is well-written and clear.

I wonder if the last sentence of the first paragraph is misleading – is the high rate of relapse an indictment of the treatments or just a commentary on the episodic nature of the illness?

Perhaps for this sentence specify that these trials have examined TFTs as stand-alone treatments specifically for depression: “An increasing number of clinical trials have been conducted investigating a range of TFTs (exposure-based cognitive therapy, EMDR, trauma-focused-CBT, and imagery rescripting) as a treatment for depression”

For the second paragraph, the authors discuss targeting the intrusive thoughts – but TFT’s target much more broadly the symptoms of PTSD (especially avoidance for some of them).

Perhaps use more specific and less colloquial language: “This was done via a systematic review of all randomised controlled trials and a meta-analysis of the pooled data to get an understanding of the overall state of the scientific evidence.” i.e. what exactly are the aims or hypotheses?

Methods:

The methods section appears reasonable.

Whether or not studies requiring a trauma were excluded is not clear.

It is unclear to me why non-clinical populations would be included in these studies? “The populations sampled were expected to be varied including clinical and non-clinical populations”

Results:

Is there an extra comma here? “751, studies were initially identified”

It is somewhat unclear to me if studies were excluded for requiring a trauma experience (different information in methods vs. results). This doesn’t seem to me like it would be a problem. If there is no trauma experience, then what would be the rationale for using a TFT?

For the studies where the TFT was an adjunct, what was the primary treatment?

The range of sessions is given (1 to 18), but perhaps a mean would also be helpful. I had to look at the tables to determine that the 1-session treatment is by design and not simply due to dropout; a one-session TFT seems qualitatively different than most substantive TFTs.

This seems hard to believe: “All three trials that compared TFT to an active control used EMDR as the TFT and CBT (non-trauma-focused) as the active control intervention (n = 129). The mean effect size of this analysis was moderate [d = 0.66 (95% CI: 0.31� 1.02)] with zero heterogeneity in favour of the TFT.” Was there an allegiance effect in these studies? How to explain this?

Did any of the studies report on PTSD symptoms (in the absence of a PTSD diagnosis)? In the discussion the authors note that these changes are seen “outside of a PTSD diagnosis” – I think one question is whether these patients may have had an (undiagnosed) PTSD diagnosis or symptoms.

Discussion:

Change “deigns” to designs.

I don’t think this sentence is supported by the available evidence: “Therefore, the evidence that there is an additional psychological intervention that is at least as effective as CBT but possibly targets different mechanisms or processes is welcomed.”

Overall, the discussion seems to be mostly a review of the results with little interpretation.

As a future direction, it would seem to me that it would be useful to test whether there are certain sub-populations that respond better to TFT. One would imagine that these would be most useful for those for whom an adverse event/trauma is part of the etiological picture of depression – otherwise it is theoretically unclear why a TFT would be used. Indeed, I would like to see the authors address this question in the discussion of why a TFT should indeed be expected to have an effect on depression in the absence of PTSD. What might be the mechanisms of treatment? Why would one choose a TFT over another front-line treatment for depression? Why use a TFT versus something like schema therapy or an approach that is geared toward adversities rather than traumas per se? If more rigorous studies found similar results to these, what would that tell us about the etiology of depression? Or about trans-diagnostic mechanisms of change in treatment? Or the overlap among stressful-event and trauma-related symptoms?

It is notable that EMDR was the TFT in all but one study. Why might that be? Is it perhaps that it is the least “trauma-focused” of the TFT’s? Prolonged exposure would entail a significant amount of time addressing the trauma itself, and Cognitive Processing Therapy has significant overlap with cognitive therapy for depression.

It is also notable that bias rates are quite high for this group of studies (along with small samples and almost universal self-report measures), which dampens my enthusiasm (or at least trust in) the results, and I think should be further highlighted as a limitation – not of the paper but of the literature.

6. PLOS authors have the option to publish the peer review history of their article (what does this mean?). If published, this will include your full peer review and any attached files.

Reviewer #1: No

Reviewer #2: No

---

## [Author Response · Author response to Decision Letter 0]

28 Apr 2021

Response to Reviewers

Thank you for taking the time to review our article "Trauma Focused Treatments for Depression. A Systematic Review and Meta-Analysis". We are grateful for the feedback provided and believe that this has substantially improved the quality of our paper. We have responded to these comments below and attached an updated manuscript highlighting changes made and responding to the specific points raised. 

Reviewer #1: Thank you for the opportunity to review this manuscript reviewing trauma-focused treatments for depression. While I think the intention of the review would be of interest to readers, the frame of "trauma-focused treatments" seems misleading (discussed below). The authors conclude that EMDR would be a beneficial treatment for depression. 

1. This should be emphasized and conjectures about the use of TFTs more broadly should be removed, especially in the discussion section. 

Thank you, this is an important point, and we have considered it in detail. When we set out to do the review, we aimed to look at TFTs in general. We were aware of clinicians/ researchers from different approaches to trauma treatments advocating that TFTs for depression was efficacious (Hayes et al., 2007; Minelli et al., 2019; Moritz et al., 2018), including a book dedicated to imagery-based cognitive therapy on mood disorders in which in chapter 9 the authors advocate for imagery rescripting as a treatment for depression (Holmes, Hales, Young, & Di Simplicio, 2019). Thus, in registering our review with PROSPERO we stated what we intended to search, which is TFTs in all shapes and forms. Therefore, we believed it would not be objective scientific reporting to change the paper's title. Our rationale was to look at TFTs in general. We found that despite several outcome studies using TFTs for depression, there was a dearth of TFTs other than EMDR being tested using RCT designs. We agree with the reviewer that this is a significant point and important to make clear in the manuscript. Thus, we have articulated this story in the revised manuscript and discussed the need to have more RCTs on TFTs other than EMDR to further test the hypothesis. 

2. Generally, highlighting in the introduction mechanisms of change underlying various TFTs will be important to add, as this provides theoretical rationale for why these treatments might be beneficial for individuals suffering from depression. 

We have expanded the introduction to highlight the role of adverse events in predisposing individuals to depression and the high frequency of intrusive memories for depressed individuals. We also hypothesise the role of these experiences and related intrusions in maintenance for ongoing depression. 

3. In the discussion, discussing why EMDR is the only TFT used for depression in the studies reviewed is also important to examine further. Overall, this paper makes a contribution to the field, but significant changes/additions to the introduction and discussion are needed.

Thank you for this comment. Yes, we were surprised that only one study of the 11 RCTs was not EMDR using our criteria. This is particularly surprising as most PTSD treatments are as effective at decreasing comorbid depression symptoms as they are at decreasing PTSD symptoms (Ronconi et al 2015). Also, there are several non-randomised outcome studies that have shown support for TFTs in patients with depression. These include cognitive therapy (Hayes et al., 2007), trauma-focused-CBT (Minelli et al., 2019) and a variety for imagery rescripting (Wheatley & Hackmann, 2011). In the revised manuscript, we have added additional text to the introduction as to why it makes sense to assess the evidence for TFTs in depression which draws on the role of aversive events and intrusive memories. We continue this in the discussion and include refences to schema therapy (as suggested by the other reviewers) to further provide a conceptual model that supports why TFTs for depression would be beneficial. 

4. Please provide peer-reviewed citations for the benefits of trauma-focused treatments for depression. Citing ISTSS does not support the suggestion that benefits of TFT for depression are well-documented.

Thank you for your comment. We have taken this on board and added additional references. These are Ehring et al., 2014; International Society of Traumatic Stress Studies, 2019; Lenz & Hollenbaugh, 2015; and Ronconi, Shiner, & Watts, 2015. 

5. Please fix this sentence accordingly: "An increasing number of clinical trials have been conducted investigating a range of TFTs (exposure-based cognitive therapy, EMDR, trauma-focused-CBT, and imagery rescripting) as treatments for depression."

Thank you. We have altered the sentence, and it now reads: "An increasing number of clinical trials have been conducted investigating a range of TFTs as a treatment for depression outside a PTSD diagnosis, including exposure-based cognitive therapy (Hayes et al., 2007), EMDR (Hase et al., 2015), trauma-focused-CBT (Minelli et al., 2019), and imagery rescripting (Moritz et al., 2018)". 

6. The following sentence is wordy and difficult to follow – please revise to something like, "Following the increase in published studies examining TFT for depression, a number of narrative reviews have been published highlighting the promise of such interventions."

We have revised this sentence. It is now: "Due to the number of published manuscripts relating to using a TFT for depression, narrative reviews of both EMDR (Carletto et al., 2017; Malandrone, Carletto, Hase, Hofmann, & Ostacoli, 2019) and imagery rescripting (Wheatley & Hackmann, 2011) have been published. These reviews highlight that each TFT has been found to be effective with depressed individuals in the absence of any specific PTSD type trauma background".

7. The introduction would benefit from more detailed examination of previous studies looking at TFTs for depression. What are the mechanisms of change seen across studies with different treatments? Moreover, while adverse childhood experiences are associated with depression, what factors lead to a primary diagnosis of depression over PTSD in these cases and how might this account for benefits from TFT treatment for depressed individuals? If individuals suffering from depression did not experience a significant adverse event precipitating symptom onset, would they derive the same benefits from TFTs? The intro is substantially lacking and more examination of these questions is warranted.

Thank you for your comments. We have elaborated on why TFTs may be of benefit for depression in the introduction. 

We have also made the steps of reasoning easy to follow in the introduction. That is:

i. Adverse life events are very common in depression and correlate with poor treatment response and symptom chronicity.

ii. People with depression have intrusive memories of such experiences, almost as common as people with PTSD.

iii. Many adverse events fail to make criterion A for PTSD, e.g. emotional abuse, life-threatening illness (specifically excluded in DSM 5), neglect, and bullying.

iv. There are theoretical reasons as to why intrusive memories might account for depression. 

Further, we discuss that many aversive events are not sufficient to give rise to a PTSD diagnosis including eg bullying, parental abandonment, and neglect. Nevertheless, these can be experienced as traumatic and hence can benefit from a trauma-based approach.

Method

8. Were individuals with comorbid diagnoses of PTSD and depression considered?

While the inclusion of individuals with comorbid PTSD was considered, there is already substantial evidence to highlight the efficacy of TFTs on depression for these individuals (Ehring et al., 2014; International Society of Traumatic Stress Studies, 2019; Lenz & Hollenbaugh, 2015; Ronconi, Shiner, & Watts, 2015). We have added further references in the introduction to reflect this evidence. However, our focus was to look at individuals who do not have PTSD to assess if TFTs are helpful for depressive symptoms outside the PTSD diagnosis, and thus we limited the inclusion criteria to these studies.

9. This sentence is wordy and difficult to follow. Please consider revising… "Studies with an inclusion criterion of adverse or traumatic events, including neglect or other incidents that would fall outside of the DSM 5 diagnostic criteria for PTSD, were not excluded from the analysis."

We have reworded this sentence and the one prior and given several examples for clarity. It now reads: "As the focus of our study was individuals with depression outside a PTSD diagnosis participants in the studies were not required to have a history of trauma as defined by the major diagnostic criterion (i. e. American Psychiatric Association, 2013; World Health Organisation, 2018). However, studies with an inclusion criterion of adverse or traumatic experiences that specified a broad definition of adversities to include incidents such as neglect, parental divorce, or bullying, were included in the analysis."

10. Please fix this sentence accordingly: "Analysis from baseline was conducted using post-treatment and at follow-up results, from continuous and dichotomous outcomes."

Thank you. We have altered the sentence to now reads: "Analysis of variable change from baseline was conducted using post-treatment and at follow-up results, from continuous and dichotomous outcomes." 

Results

11. Please fix this sentence accordingly: "As shown in Fig 1, 751 studies were initially identified, and 372 duplicates were removed."

We have altered this sentence for clarity to: "As shown in Fig 1, 751 studies were identified in the initial search, including 372 duplicates which were removed."

12. Please fix this sentence accordingly: "A total of 11 studies with a sum of 567 participants were eligible for analysis."

Thank you. This has been amended: "Following this, 11 studies, which involved a total of 567 participants, were considered eligible and included in the analysis."

13. Is there utility in keeping the study investigating rescripting given that the remaining studies used EMDR? Moreover, given that all studies (with one exception) used EMDR rather than other trauma-focused treatments, is it possible that this speaks to the fact that trauma treatments are not necessarily beneficial for depression but treatments aimed at alleviating distress/processing negative memories are beneficial. It seems that other trauma therapies (i.e., CPT, exposure therapy, TF-CBT) emphasize exposure/cognitive restructuring/behavioral change whereas EMDR is focused on memory reconsolidation.

Thank you. This is an important point. We agree that all trauma therapies aim to eventually alleviate the distress of negative memories (Schnyder et al., 2015). They differ on the type of exposure, the degree that they engage the client to focus on the recollection of the event, the meaning associated with the event, or the extent they directly target behaviour change. Prolonged exposure involves extensive retelling of the trauma experience and further exposure to these memories via listening to recordings of these sessions. Whilst there is diversity in opinion as to the active mechanisms of this treatment, habituation rather than memory reconsolidation has been emphasised in prolonged exposure in contrast to EMDR (Careaga, Girardi, & Suchecki, 2016). We do not think that it is the mechanism underlying the therapy that explains why so few studies met our inclusion criteria. In our view, if a treatment works on an intrusive memory of a negative event, the type of event (i.e. whether that memory is verbal bullying or a physical assault) is unlikely to make a difference. One possible reason for this has to do with the underlying theory. EMDR is taught within a theory of what Shapiro described as Adaptive Information Processing. Central to this theory is the idea that negative life events can lead to negative self-view or a negative sense of the world that can lead to psychopathology. It is not a theory of PTSD per se but, in essence, a transdiagnostic approach. Perhaps other TFT approaches have focused on PTSD symptoms, so clinicians and researchers have been less likely to consider investigating how their treatment works on non PTSD type traumas. 

We acknowledge that our response to this point from the reviewer is speculative. Because of this we decided not to include these thoughts in the revised manuscript however we could include them if they were deemed to make a contribution. 

14. Was the analysis of EMDR vs. CBT looking specifically at behavioral activation or a general CBT model? Given that BA is an intended treatment for depression, this would be an important distinction.

In the four studies that used CBT as a comparator, the intervention was labelled CBT by the writers. The interventions all included cognitive restructuring and behaviour homework tasks. None were purely behavioural activation. 

Discussion

15. The general conclusion drawn from this meta-analysis is misleading – TFTs are not necessarily a promising approach to treating depressive symptoms. EMDR may be a promising approach, according to findings. Overall, the discussion would benefit from exploration of mechanisms of change underlying EMDR for depression, and some mention of why other TFT have not been used solely for depression.

 Thank you for pointing this out. We certainly do not intend anything to be perceived as misleading. However, we do not agree with the comment that TFTs have not been used for depression when this is the sole issue. As discussed in point 5 above, TFTs have been used and advocated as treatments for depression, and we have provided the basis for this claim. In the revised manuscript, we have expanded the introduction and discussion to document this. However, our prescribed literature search meant that several outcome studies did not meet the inclusion criteria. In the discussion, we have been more circumspect about TFTs in general as applied to depression, given we only found the one non-EMDR RCT that met our criteria. We have identified this as a future research priority.

General

16. Please read through the manuscript from grammatical errors.

Thank you, this has been addressed.

 

Reviewer #2: This manuscript aims to provide a meta-analysis of studies using trauma-focused treatments to treat depression without a diagnosis of PTSD. Given that there are now 11 such studies it is worthwhile to summarize them. It seems to me that this paper would be more helpful with a bit more of a critique of the existing literature and attention to the question of why such treatments may be helpful.

Introduction:

1. The introduction is well-written and clear.

Thank you and noting your first comment we have expanded the introduction to address the question of why such treatments might be helpful.

2. I wonder if the last sentence of the first paragraph is misleading – is the high rate of relapse an indictment of the treatments or just a commentary on the episodic nature of the illness?

We have added the following additional sentence to clarify this. "Although these relapse rates may be a reflection of the episodic nature of the disorder (Kanter, Busch, Weeks, & Landes, 2008), further investigation into evidence based interventions is warranted."

3. Perhaps for this sentence specify that these trials have examined TFTs as stand-alone treatments specifically for depression: "An increasing number of clinical trials have been conducted investigating a range of TFTs (exposure-based cognitive therapy, EMDR, trauma-focused-CBT, and imagery rescripting) as a treatment for depression"

Thank you. We have altered this sentence. It now reads: "An increasing number of clinical trials have been conducted investigating a range of TFTs as a treatment for depression outside a PTSD diagnosis, including exposure-based cognitive therapy (Hayes et al., 2007), EMDR (Hase et al., 2015), trauma-focused-CBT (Minelli et al., 2019), and imagery rescripting (Moritz et al., 2018)." 

4. For the second paragraph, the authors discuss targeting the intrusive thoughts – but TFT's target much more broadly the symptoms of PTSD (especially avoidance for some of them).

Thank you, we have altered the introduction to include experiential avoidance, intrusive images, and emotions.

5. Perhaps use more specific and less colloquial language: "This was done via a systematic review of all randomised controlled trials and a meta-analysis of the pooled data to get an understanding of the overall state of the scientific evidence." i.e. what exactly are the aims or hypotheses?

This has been altered. It now reads: "It was hypothesised that, for individuals with depression as their primary complaint, those who receive a TFTs would be more likely to show a decrease in depressive symptoms and increase the likelihood of remission than those who receive a control condition. This hypothesis was assessed via a systematic review of all relevant randomised controlled trials (RCTs) and a meta-analysis of the pooled data."

Methods:

6. The methods section appears reasonable.

Thank you.

7. Whether or not studies requiring a trauma were excluded is not clear.

We have elaborated on this point for clarity.

8. It is unclear to me why non-clinical populations would be included in these studies? "The populations sampled were expected to be varied including clinical and non-clinical populations".

Thank you for pointing this out. This was a misleading sentence. In the revised manuscript the sentence now reads: "The studies were expected to vary in terms of number of sessions, type of TFT and the clinical severity of the participants, therefore a random effects model was used in all analyses." The fact that one study only included a structured clinical interview is now discussed under limitations in the discussion. 

Results:

9. Is there an extra comma here? "751, studies were initially identified"

Thank you, this has been removed.

10. It is somewhat unclear to me if studies were excluded for requiring a trauma experience (different information in methods vs. results). This doesn't seem to me like it would be a problem. If there is no trauma experience, then what would be the rationale for using a TFT?

We have made this more explicit in the introduction. A trauma experience does not have to be one that could potentially give rise to PTSD. One of the main tenants of this paper focuses on adverse events, which is a broader category. Such events include 'trauma of omissions' (not having core needs met such as emotional or physical neglect), abuse experiences that do not involve a threat to life, such as verbal abuse from a childhood carer or bullying from peers. 

11. For the studies where the TFT was an adjunct, what was the primary treatment?

We have updated this in the results section. The adjunct psychotherapy used in two of these trials was delivered in groups and based on a CBT or psychodynamic model (Dominguez, Drummond, Gouldthorp, Janson, & Lee, 2020; Hase et al., 2018), and one involved individual CBT (Hogan, 2002). 

12. The range of sessions is given (1 to 18), but perhaps a mean would also be helpful. I had to look at the tables to determine that the 1-session treatment is by design and not simply due to dropout; a one-session TFT seems qualitatively different than most substantive TFTs.

Thank you. We have amended the results to clarify the number of sessions described in the table as per the intervention design. We have also including the average number of sessions. It now reads: "In the imagery rescripting study the intervention was self-administered, and the number of therapy sessions were not recorded. For the remaining trials number of TFT sessions in the studies' designs ranged from 1 to 18 (average 6.5) and were between 45 to 120 minutes in duration."

13. This seems hard to believe: "All three trials that compared TFT to an active control used EMDR as the TFT and CBT (non-trauma-focused) as the active control intervention (n = 129). The mean effect size of this analysis was moderate [d = 0.66 (95% CI: 0.31~ 1.02)] with zero heterogeneity in favour of the TFT." Was there an allegiance effect in these studies? How to explain this?

This was a surprise to us as well. It may be impacted by the quality of the included studies as demonstrated in the risk of bias analysis, given two of the included studies had some or high risk of bias. However, as mentioned more generally in the discussion, most of the data in this analysis was self-report, which could influence results. Hopefully, these findings lead to further testing of the interventions.

14. Did any of the studies report on PTSD symptoms (in the absence of a PTSD diagnosis)? In the discussion the authors note that these changes are seen "outside of a PTSD diagnosis" – I think one question is whether these patients may have had an (undiagnosed) PTSD diagnosis or symptoms.

None of the studies specifically assessed for and excluded individuals with PTSD or reported on PTSD symptoms for all participants. This has been identified in the discussion and highlighted as an important area for future research. 

Discussion:

15. Change "deigns" to designs.

This has been altered.

16. I don't think this sentence is supported by the available evidence: "Therefore, the evidence that there is an additional psychological intervention that is at least as effective as CBT but possibly targets different mechanisms or processes is welcomed."

We have altered this sentence. It now reads: "Therefore, the evidence that there is an additional psychological intervention that is at least as effective as CBT but possibly targets different mechanisms or maintaining factors, such as intrusive memories, is welcomed."

17. Overall, the discussion seems to be mostly a review of the results with little interpretation.

We have extended several key points in the discussion regarding the relevance and implications for future research and clinical practice.

18. As a future direction, it would seem to me that it would be useful to test whether there are certain sub-populations that respond better to TFT. One would imagine that these would be most useful for those for whom an adverse event/trauma is part of the etiological picture of depression – otherwise it is theoretically unclear why a TFT would be used. Indeed, I would like to see the authors address this question in the discussion of why a TFT should indeed be expected to have an effect on depression in the absence of PTSD. What might be the mechanisms of treatment? Why would one choose a TFT over another front-line treatment for depression? Why use a TFT versus something like schema therapy or an approach that is geared toward adversities rather than traumas per se? If more rigorous studies found similar results to these, what would that tell us about the etiology of depression? Or about trans-diagnostic mechanisms of change in treatment? Or the overlap among stressful-event and trauma-related symptoms?

Thank you for your comment. We have considerably altered the introduction to argue why TFTs would have an effect on depression. As described in point 7 of the other reviewers comments the argument is:

i. Adverse life events are very common in depression and correlate with poor treatment response and symptom chronicity.

ii. People with depression have intrusive memories of such experiences almost as frequently as people with PTSD.

iii. Many adverse events fail to make criterion A for PTSD, e.g. emotional abuse, life threatening illness (specifically excluded in DSM 5), neglect, and bullying.

iv. There are theoretical reasons as to why intrusive memories might account for depression. 

In the discussion, we continue to examine these issues and further illustrate theories as to why TFTs are indicated for depression using schema therapy as you suggest. We added: "The focus on adverse events as aetiologically related to current psychopathology is central to schema therapy (Young, Klosko, & Weishaar, 2003). In schema theory the experiences of having core needs not met are viewed as the basis of what Young called early maladaptive schemas that then cause dysfunction throughout the person's life. In schema therapy these experiences are targeted with interventions such as imagery rescripting, chair work, or EMDR (Young, Zangwill, & Behary, 2002)." 

19. It is notable that EMDR was the TFT in all but one study. Why might that be? Is it perhaps that it is the least "trauma-focused" of the TFT's? Prolonged exposure would entail a significant amount of time addressing the trauma itself, and Cognitive Processing Therapy has significant overlap with cognitive therapy for depression.

The fact that there was only one relevant non-EMDR RCT was a surprise to us and highlighted a significant gap in the literature as we identify in the discussion. One hypothesis is that several other TFTs, including prolonged exposure and cognitive processing therapy, have similar origins to many evidence-based depression treatments, coming from a cognitive or behavioural theoretical background. EMDR and imagery rescripting are both relatively new interventions and based on differing theoretical backgrounds. Thus, it may be considered that there is more interest in exploring the impact of these newer interventions, with varied theoretical background. However, this is purely speculation, and it is outside of the scope of this study to adequately address why past researchers have chosen to use particular interventions over others. 

20. It is also notable that bias rates are quite high for this group of studies (along with small samples and almost universal self-report measures), which dampens my enthusiasm (or at least trust in) the results, and I think should be further highlighted as a limitation – not of the paper but of the literature.

Thank you, this seems valid. We have elaborated on this point in the discussion on page 20.

 

References

American Psychiatric Association. (2013). Diagnostic and statistical manual of mental disorders: DSM-5: Washington, DC: American psychiatric association.

Careaga, M. B. L., Girardi, C. E. N., & Suchecki, D. (2016). Understanding posttraumatic stress disorder through fear conditioning, extinction and reconsolidation. Neuroscience & Biobehavioral Reviews, 71, 48-57. 

Carletto, S., Ostacoli, L., Colombi, N., Luca, C., Oliva, F., Isabel, F., & Arne, H. (2017). EMDR for depression: A systematic review of controlled studies. 

Dominguez, S. K., Drummond, P., Gouldthorp, B., Janson, D., & Lee, C. W. (2020). A randomized controlled trial examining the impact of individual trauma‐focused therapy for individuals receiving group treatment for depression. Psychology and Psychotherapy: Theory, Research and Practice. 

Ehring, T., Welboren, R., Morina, N., Wicherts, J. M., Freitag, J., & Emmelkamp, P. M. (2014). Meta-analysis of psychological treatments for posttraumatic stress disorder in adult survivors of childhood abuse. Clinical Psychology Review, 34(8), 645-657. 

Hase, M., Balmaceda, U. M., Hase, A., Lehnung, M., Tumani, V., Huchzermeier, C., & Hofmann, A. (2015). Eye movement desensitization and reprocessing (EMDR) therapy in the treatment of depression: a matched pairs study in an inpatient setting. Brain and Behavior, 5(6). 

Hase, M., Plagge, J., Hase, A., Braas, R., Ostacoli, L., Hofmann, A., & Huchzermeier, C. (2018). Eye movement desensitization and reprocessing versus treatment as usual in the treatment of depression: a randomized-controlled trial. Frontiers in psychology, 9. 

Hayes, A. M., Feldman, G. C., Beevers, C. G., Laurenceau, J.-P., Cardaciotto, L., & Lewis-Smith, J. (2007). Discontinuities and cognitive changes in an exposure-based cognitive therapy for depression. Journal of consulting and clinical psychology, 75(3), 409. 

Hogan, W. A. (2002). The comparative effects of eye movement desensitization and reprocessing (EMDR) and cognitive behavioral therapy (CBT) in the treatment of depression. 

Holmes, E. A., Hales, S. A., Young, K., & Di Simplicio, M. (2019). Imagery-based cognitive therapy for bipolar disorder and mood instability: Guilford Publications.

International Society of Traumatic Stress Studies. (2019). Posttraumatic Stress Disorder. Prevention and Treatment Guidelines. Retrieved from www.istss.org/treating-trauma/new-istss-prevention-and-treatment-guidelines 

Kanter, J. W., Busch, A. M., Weeks, C. E., & Landes, S. J. (2008). The nature of clinical depression: Symptoms, syndromes, and behavior analysis. The Behavior Analyst, 31(1), 1-21. 

Lenz, A. S., & Hollenbaugh, K. M. (2015). Meta-analysis of trauma-focused cognitive behavioral therapy for treating PTSD and co-occurring depression among children and adolescents. Counseling Outcome Research and Evaluation, 6(1), 18-32. 

Malandrone, F., Carletto, S., Hase, M., Hofmann, A., & Ostacoli, L. (2019). A brief narrative summary of randomized controlled trials investigating EMDR treatment of patients with depression. Journal of EMDR Practice and Research, 13(4), 302-306. 

Minelli, A., Zampieri, E., Sacco, C., Bazzanella, R., Mezzetti, N., Tessari, E., . . . Bortolomasi, M. (2019). Clinical efficacy of trauma-focused psychotherapies in treatment-resistant depression (TRD) in-patients: A randomized, controlled pilot-study. Psychiatry Research, 273, 567-574. doi:http://dx.doi.org/10.1016/j.psychres.2019.01.070

Moritz, S., Ahlf-Schumacher, J., Hottenrott, B., Peter, U., Franck, S., Schnell, T., . . . Jelinek, L. (2018). We cannot change the past, but we can change its meaning. A randomized controlled trial on the effects of self-help imagery rescripting on depression. Behaviour Research and Therapy, 104, 74-83. 

Ronconi, J. M., Shiner, B., & Watts, B. V. (2015). A meta-analysis of depressive symptom outcomes in randomized, controlled trials for PTSD. The Journal of nervous and mental disease, 203(7), 522-529. 

Schnyder, U., Ehlers, A., Elbert, T., Foa, E. B., Gersons, B. P., Resick, P. A., . . . Cloitre, M. (2015). Psychotherapies for PTSD: what do they have in common? European journal of psychotraumatology, 6(1), 28186. 

Wheatley, J., & Hackmann, A. (2011). Using imagery rescripting to treat major depression: theory and practice. Cognitive and Behavioral Practice, 18(4), 444-453. 

World Health Organisation. (2018). International classification of diseases for mortality and morbidity statistics (11th Revision).

Young, J. E., Klosko, J. S., & Weishaar, M. E. (2003). Schema therapy: A practitioner's guide. New York: Guilford Press.

Young, J., Zangwill, W. M., & Behary, W. E. (2002). Combining EMDR and schema-focused therapy: The whole may be greater than the sum of the parts.

---

## [Decision Letter · Decision Letter 1]

8 Jun 2021

PONE-D-20-24791R1

Trauma Focused Treatments for Depression.  A Systematic Review and Meta-Analysis

PLOS ONE

Dear Dr. Lee,

Thank you for submitting your manuscript to PLOS ONE. After careful consideration, we feel that it has merit but does not fully meet PLOS ONE’s publication criteria as it currently stands. Therefore, we invite you to submit a revised version of the manuscript that addresses the points raised during the review process.

Besides fixing some typos and grammar problems, both reviewers request a clearer discussion of the limits of the technique and a better clarification of the range of condition considered (marking the difference between PTSD and primary depression). Please address both reviewers comments in your re-revised version.

We look forward to receiving your revised manuscript.

Kind regards,

Andrea Martinuzzi

Academic Editor

PLOS ONE

Journal Requirements:

Reviewers' comments:

Reviewer's Responses to Questions

**Comments to the Author**

1. If the authors have adequately addressed your comments raised in a previous round of review and you feel that this manuscript is now acceptable for publication, you may indicate that here to bypass the “Comments to the Author” section, enter your conflict of interest statement in the “Confidential to Editor” section, and submit your "Accept" recommendation.

Reviewer #1: (No Response)

Reviewer #2: (No Response)

2. Is the manuscript technically sound, and do the data support the conclusions?

Reviewer #1: Partly

Reviewer #2: Yes

3. Has the statistical analysis been performed appropriately and rigorously? 

Reviewer #1: Yes

Reviewer #2: Yes

4. Have the authors made all data underlying the findings in their manuscript fully available?

Reviewer #1: Yes

Reviewer #2: Yes

5. Is the manuscript presented in an intelligible fashion and written in standard English?

Reviewer #1: Yes

Reviewer #2: Yes

6. Review Comments to the Author

Reviewer #1: I appreciate the responses to the original comments. I still have concerns that the current data does not fully support the claim that TFTs are an effective treatment for depression, given that aside from one study only EMDR is examined. Additionally, the way that depression is being discussed does not fully capture the diagnosis or depression symptoms outside of adversity exposure, which does not occur for all individuals who experience depression. I have expanded on these concerns below.

The types of adversity you are highlighting as leading to depression (e.g., neglect, parental divorce, bullying) are those that are often discussed in research on developmental trauma disorder or complex trauma, proposed diagnoses that are meant to capture exposure to chronic trauma, particularly in childhood (see Pearlman & Courtois, 2005; Van der Kolk et al., 2009). Moreover, causes of depression are not always adversity-based, further highlighting a need to soften these claims. They can be biological, related to lack of achievement/purpose, loss of intimacy, feeling out of control. Individuals with a history of adversity may have more severe symptoms, as you mentioned, and these adversities may not lead to a PTSD diagnosis based on the limitations of the PTSD criteria, but again this is not representative of major depression as a diagnosis - this is representative of the experience of a subset of individuals whose symptoms may actually be captured by a complex trauma diagnosis.

I would like to see some discussion of rumination in the context of your reference to intrusive memories, similar to how you reference avoidance. Rumination is a core feature of depression that often occurs as a strategy to manage intrusive memories and contributes to the frequency/maintenance of intrusive memories. This would support your argument further. Moreover, I would recommend including the review by Payne et al. (2019) to add to the support of the associations between intrusive memories and depression, given that this is more recent than the Wheatley & Hackman (2011) citation and offers some language around why trauma-focused treatments such as EMDR may be beneficial, while also acknowledging limitations.

Though I continue to think the language around TFTs being beneficial for depression should be tempered, I ultimately defer to the editor as I think the findings do make a contribution to the field and highlight the need to further explore TFTs as a treatment for depression. I would, however, like to see the recommendations in the paragraph above addressed, as I think a discussion of rumination in the context of depression/intrusive memories is important (the recent meta-analysis by Mihailova & Jobson, 2018 may be useful) and the updated Payne et al. cite/examination of those findings will add to the paper.

Reviewer #2: Please do a read-through for typos/grammar (e.g. “defied” instead of “defined).

I think it would be helpful to clarify further in the “Selection Criteria” section that participant studies did not require a diagnosis of PTSD (or required NO diagnosis of PTSD?).

I agree with the first reviewer that it still reads to me as a bit misleading in the first and last paragraphs of the discussion to state that TFTs are promising (when really we’re just talking about EMDR). While I understand that the authors set out to examine TFTs (so it makes sense to still label the paper and hypotheses in that way), it seems there should be a middle ground in the discussion that better represents what was found.

It may make sense to mention that if we are examining TFTs for adverse events, that perhaps it no longer makes sense to refer to them as TFT’s (perhaps adversity-orientated treatments?).

One differential between PTSD and depression is “intrusive” thoughts vs “ruminative thoughts”. This made me wonder if depression studies have found that people with depression experience thoughts that are truly “intrusive” in the way defined by PTSD.

7. PLOS authors have the option to publish the peer review history of their article (what does this mean?). If published, this will include your full peer review and any attached files.

Reviewer #1: No

Reviewer #2: No

---

## [Author Response · Author response to Decision Letter 1]

1 Jul 2021

Response to reviewers

Thank you for your comments, we have taken on board your suggestions and believe that this has improved the quality of this manuscript. We have addressed your specific comments below. In addition, we have added the following references:

Birrer, E., & Michael, T. (2011). Rumination in PTSD as well as in traumatized and non-traumatized depressed patients: A cross-sectional clinical study. Behavioural and cognitive psychotherapy, 39(4), 381-397. 

Cowdrey, F. A., & Park, R. J. (2012). The role of experiential avoidance, rumination and mindfulness in eating disorders. Eating behaviors, 13(2), 100-105. 

Dickson, K. S., Ciesla, J. A., & Reilly, L. C. (2012). Rumination, worry, cognitive avoidance, and behavioral avoidance: Examination of temporal effects. Behavior therapy, 43(3), 629-640. 

Mihailova, S., & Jobson, L. (2018). Association between intrusive negative autobiographical memories and depression: A meta‐analytic investigation. Clinical psychology & psychotherapy, 25(4), 509-524. 

Newby, J. M., & Moulds, M. L. (2010). Negative intrusive memories in depression: The role of maladaptive appraisals and safety behaviours. Journal of affective disorders, 126(1-2), 147-154. 

Payne, A., Kralj, A., Young, J., & Meiser-Stedman, R. (2019). The prevalence of intrusive memories in adult depression: A meta-analysis. Journal of affective disorders, 253, 193-202. 

And removed:

Hofmann, A., Hilgers, A., Lehnung, M., Liebermann, P., Ostacoli, L., Schneider, W., & Hase, M. (2014). Eye Movement Desensitization and Reprocessing as an Adjunctive Treatment of Unipolar Depression: A Controlled Study. Journal of EMDR Practice and Research, 8(3), 103-112. 

Holmes, E. A., Blackwell, S. E., Heyes, S. B., Renner, F., & Raes, F. (2016). Mental imagery in depression: Phenomenology, potential mechanisms, and treatment implications. Annu Rev Clin Psychol, 12. 

Reviewer #1: 

I appreciate the responses to the original comments. I still have concerns that the current data does not fully support the claim that TFTs are an effective treatment for depression, given that aside from one study only EMDR is examined. Additionally, the way that depression is being discussed does not fully capture the diagnosis or depression symptoms outside of adversity exposure, which does not occur for all individuals who experience depression. I have expanded on these concerns below.

1. The types of adversity you are highlighting as leading to depression (e.g., neglect, parental divorce, bullying) are those that are often discussed in research on developmental trauma disorder or complex trauma, proposed diagnoses that are meant to capture exposure to chronic trauma, particularly in childhood (see Pearlman & Courtois, 2005; Van der Kolk et al., 2009). Moreover, causes of depression are not always adversity-based, further highlighting a need to soften these claims. They can be biological, related to lack of achievement/purpose, loss of intimacy, feeling out of control. Individuals with a history of adversity may have more severe symptoms, as you mentioned, and these adversities may not lead to a PTSD diagnosis based on the limitations of the PTSD criteria, but again this is not representative of major depression as a diagnosis - this is representative of the experience of a subset of individuals whose symptoms may actually be captured by a complex trauma diagnosis.

Thank you for your comment. In the paper we cite several sources that identify the well-established link between adversities and depression (e. g. Laugharne, Lillee, & Janca, 2010; Nanni, Uher, & Danese, 2012; Risch et al., 2009). While we are not proposing that this is the only factor that impacts on the likelihood of depression or the severity of diagnosis, we are proposing that, similar to biological factors, it is a significant predisposing and maintaining factor for many who have the disorder, and one that is largely neglected in depression interventions. In patients with a trauma-related disorder the characteristics of the traumatic event itself can be different than in patients with depression. In the first case the event has to be related to actual or threat of death, physical or sexual violence, but in the case of depression these events can have a totally different character. In a previous paper treated people with depression without soliciting whether they had a history of adverse events or not (Dominguez, Drummond, Gouldthorp, Janson, & Lee, 2020). The types of traumas that were treated are detailed in the slide below (Dominguez, 2021). In this study we were not treating those patients who would make criteria for complex trauma. These patients would have needed many more sessions to achieve remission of their depression diagnosis. In our revised submission we have elaborated on the prevalence of intrusive memories, and related avoidance and rumination in depressed individuals and the role of these in symptom exacerbation and maintenance as suggested. 

2. I would like to see some discussion of rumination in the context of your reference to intrusive memories, similar to how you reference avoidance. Rumination is a core feature of depression that often occurs as a strategy to manage intrusive memories and contributes to the frequency/maintenance of intrusive memories. This would support your argument further. Moreover, I would recommend including the review by Payne et al. (2019) to add to the support of the associations between intrusive memories and depression, given that this is more recent than the Wheatley & Hackman (2011) citation and offers some language around why trauma-focused treatments such as EMDR may be beneficial, while also acknowledging limitations.

Thank you for this comment. We have altered the introduction by updating the information from the Payne et al. (2019) and Mihailova & Jobson (2018) paper and also added information regarding rumination.

3. Though I continue to think the language around TFTs being beneficial for depression should be tempered, I ultimately defer to the editor as I think the findings do make a contribution to the field and highlight the need to further explore TFTs as a treatment for depression. I would, however, like to see the recommendations in the paragraph above addressed, as I think a discussion of rumination in the context of depression/intrusive memories is important (the recent meta-analysis by Mihailova & Jobson, 2018 may be useful) and the updated Payne et al. cite/examination of those findings will add to the paper.

As stated above we have included material on rumination and the references you provided which strengthened the paper. We have further altered the abstract and discussion to be more circumspect around the evidence for TFT in general. 

Reviewer #2: 

1. Please do a read-through for typos/grammar (e.g. “defied” instead of “defined).

Thank you. This has been addressed. 

2. I think it would be helpful to clarify further in the “Selection Criteria” section that participant studies did not require a diagnosis of PTSD (or required NO diagnosis of PTSD?).

We have added the following sentence into the PICO for clarity: “Studies that required participants to have a PTSD diagnosis were excluded from our analysis.”

3. I agree with the first reviewer that it still reads to me as a bit misleading in the first and last paragraphs of the discussion to state that TFTs are promising (when really we’re just talking about EMDR). While I understand that the authors set out to examine TFTs (so it makes sense to still label the paper and hypotheses in that way), it seems there should be a middle ground in the discussion that better represents what was found.

We have altered both the abstract and discussion to ensure this is clearer.

4. It may make sense to mention that if we are examining TFTs for adverse events, that perhaps it no longer makes sense to refer to them as TFT’s (perhaps adversity-orientated treatments?).

This is an interesting idea and our research group discussed this at length. We come to the decision that trauma therapy are common terms used both in the literature and clinical practice and that we can something that neglect, abuse etc are traumatic and that treatments that target anything that is traumatic can be called trauma focused treatment. In thinking through this issue and what is generally meant by the terms we did come across several clinical discussion sites where the concept of possibly two main categories of trauma commonly referred to as Big “T” and little “t.” Big “T” traumas are the events most commonly associated with post-traumatic stress disorder (PTSD) including serious injury, sexual violence, or life-threatening experiences. Little t traumas are any life events experienced by the individual as traumatic but would not fit the PTSD category A diagnostic criteria. We think that the treatment of each should be the same. We thought about writing this up in the discussion but decided it appeared outside the scope of the paper. However we would do this if the editor felt otherwise. 

5. One differential between PTSD and depression is “intrusive” thoughts vs “ruminative thoughts”. This made me wonder if depression studies have found that people with depression experience thoughts that are truly “intrusive” in the way defined by PTSD.

We expanded our information on intrusive memories to further specify related avoidance and rumination. We also added a sentence in the discussion to identify that future work in this area is needed.

 

References

Dominguez, S. (2021). EMDR for depression. Paper presented at the EMDR Europe Research & Practice Conference, Virtual.

Dominguez, S., Drummond, P., Gouldthorp, B., Janson, D., & Lee, C. W. (2020). A randomized controlled trial examining the impact of individual trauma‐focused therapy for individuals receiving group treatment for depression. Psychology and Psychotherapy: Theory, Research and Practice. doi:10.1111/papt.12268

Laugharne, J., Lillee, A., & Janca, A. (2010). Role of psychological trauma in the cause and treatment of anxiety and depressive disorders. Current Opinion in Psychiatry, 23(1), 25-29. 

Nanni, V., Uher, R., & Danese, A. (2012). Childhood maltreatment predicts unfavorable course of illness and treatment outcome in depression: a meta-analysis. American Journal of Psychiatry, 169(2), 141-151. 

Risch, N., Herrell, R., Lehner, T., Liang, K.-Y., Eaves, L., Hoh, J., . . . Merikangas, K. R. (2009). Interaction between the serotonin transporter gene (5-HTTLPR), stressful life events, and risk of depression: a meta-analysis. Jama, 301(23), 2462-2471.

---

## [Editor Report · Decision Letter 2]

5 Jul 2021

Trauma Focused Treatments for Depression:  A Systematic Review and Meta-Analysis

PONE-D-20-24791R2

Dear Dr. Lee,

We’re pleased to inform you that your manuscript has been judged scientifically suitable for publication and will be formally accepted for publication once it meets all outstanding technical requirements.

Kind regards,

Andrea Martinuzzi

Academic Editor

PLOS ONE
---

## [Editor Report · Acceptance letter]

14 Jul 2021

PONE-D-20-24791R2 

Trauma-Focused Treatments for Depression.  A Systematic Review and Meta-Analysis 

Dear Dr. Lee:

I'm pleased to inform you that your manuscript has been deemed suitable for publication in PLOS ONE. Congratulations! Your manuscript is now with our production department. 

Kind regards, 

on behalf of

Dr. Andrea Martinuzzi 

Academic Editor

PLOS ONE